# Expanding the Genetic and Clinical Spectrum of Hereditary Transthyretin Amyloidosis: The Glu61Ala Variant

**DOI:** 10.3390/jpm15020061

**Published:** 2025-02-06

**Authors:** Christian Messina, Salvatore Gulizia, Federica Scalia, Eugenia Borgione, Francesco Cappello, Filippo Brighina, Vincenzo Di Stefano

**Affiliations:** 1Department of Biomedicine, Neuroscience and Advanced Diagnostics (BiND), University of Palermo, 90127 Palermo, Italy; chry.messina@gmail.com (C.M.); federica.scalia02@unipa.it (F.S.); francesco.cappello@unipa.it (F.C.); filippo.brighina@unipa.it (F.B.); 2Azienda Sanitaria Provinciale di Siracusa (ASP Siracusa), Corso Gelone 17, 96100 Siracusa, Italy; dr.salvogulizia@gmail.com; 3Oasi Research Institute-IRCCS, Via Conte Ruggero 73, 94018 Troina, Italy; eborgione@oasi.en.it

**Keywords:** Glu81Ala, hereditary transthyretin amyloidosis, polyneuropathy, TTR

## Abstract

**Introduction.** Hereditary transthyretin amyloidosis (hATTR) is a rare disorder with a largely variable worldwide prevalence, and it is caused by autosomal dominant mutations in the transthyretin (*TTR*) gene, leading to cardiological, neurological, or mixed phenotypes. Apart from the Glu89Gln, Phe64Leu, and Thr49Ala variants, recently, other mutations of *TTR* gene have been reported in Sicily (His90Asn, Val122Ile, Ser77Phe, Val20Ala). With this paper, we describe a novel mutation in the *TTR* gene, the Glu61Ala variant, which had been previously reported in only one case with a cardiac phenotype, and the clinical findings surrounding it. **Materials and Methods.** One individual affected by chronic idiopathic polyneuropathy and a major red flag for hATTR underwent genetic testing to look for mutations in the *TTR* gene. Then, his relatives were subjected to the same test. We assessed the anamnestic profile and conducted general and neurological examination, blood tests, nerve conduction studies (NCS), electrocardiogram, and Sudoscan for each patient. Written informed consent was acquired for every patient. **Results.** Among 7 patients screened, 5 patients carried the Glu61Ala variant (71%). The mean age was 64.6 ± 10.2 years, whereas the mean age at onset was 59.4 ± 7.9 years. In our study, three patients (60%) showed a mixed phenotype, whereas two of them (40%) showed a neurological phenotype. **Discussion.** The Glu61Ala variant was reported only in one case with a cardiological phenotype, but our patients showed both neurological and cardiological involvement. Further studies are needed to improve knowledge of this genetic variant.

## 1. Introduction

Hereditary transthyretin amyloidosis (hATTR) is a rare disorder with a largely variable worldwide prevalence, and it is caused by autosomal dominant mutations in the transthyretin (TTR) gene, located on chromosome 18 [1]. TTR is a protein synthesized in the liver, pancreas, choroid plexus, and retinal pigment epithelium, which has the primary function of transporting the thyroid hormone thyroxine in the plasma and cerebrospinal fluid [1]. Pathogenesis of hATTR is due to the aggregation of misfolded proteins causing extracellular deposition of amyloid-insoluble fibrils and local tissue damage [1]. Unfortunately, the diagnosis of hATTR is challenging, and it can be delayed for many years, leading to disability and mortality. It can present with length-dependent sensori-motor axonal polyneuropathy (PN-hATTR), and/or infiltrative cardiomyopathy (CM-hATTR), variably associated with autonomic dysfunction, juvenile ocular disorders, and kidney disease [1]. Hence, clinical phenotypes are heterogeneous and subtle, often leading to very difficult diagnoses. The global prevalence of hATTR is estimated to be around 1/450,000 people (range 1/120,000–830,000), with a total number of confirmed cases of 10,186 people [2]. More than 150 mutations in the *TTR* gene have been described, and every mutation shows a distinct phenotype, including variable heart and nerve involvement, age of onset, disease progression, and mortality [2,3]. The Val30Met variant of the *TTR* gene was the first to be identified and is considered the most frequent in both endemic and non-endemic areas, being responsible for about 70% of hATTR [1,3]. In 2015, Mazzeo et al. reported a hATTR prevalence of 8.8/1,000,000 in Sicily; interestingly, the authors did not report the common Val30Met variant, describing only three *TTR* variants (Glu89Gln, Phe64Leu, Thr49Ala) [4]. However, recently, four further mutations of the *TTR* gene have been reported in Sicily (His90Asn, Val122Ile, Ser77Phe, Val20Ala) [5,6]. In general, correlations between genotype and phenotype in hATTR are often difficult due to the heterogeneity of clinical onset, progression, and prognosis. With this paper, we describe a family carrying the Glu61Ala variant, which had been previously reported only once, in a case of CM-hATTR [7].

## 2. Materials and Methods

At first, we identified one individual presenting at our neuromuscular clinic with chronic idiopathic polyneuropathy and at least one major red flag according to Adams et al [8]. Then, the proband was genetically, clinically, and instrumentally evaluated. As a consequence, his relatives were subjected to cascading screening and to the same evaluation in the Department of Biomedicine, Neuroscience and Advanced Diagnostic (BIND), University of Palermo. Only 6 relatives were screened. Each patient underwent diagnostic genetic testing on salivary samples to look for mutations in the *TTR* gene. They were assessed with anamnestic profiles, general and neurological examinations, blood tests, nerve conduction velocity studies (NCS), electrocardiograms, and Sudoscan.

### 2.1. Clinical Examination

Gait, cranial nerves and cerebellar function, Romberg sign, superficial and deep sensitivity, proximal and distal muscle strength, and osteotendinous reflexes (OTR) in the four limbs were evaluated. Moreover, during clinical examinations, we determined the Compound Autonomic Dysfunction Test (CADT) [9], Medical Research Council (MRC) [10], Familial Amyloidosis Polyneuropathy (FAP), Karnofsky Performance Status Scale (KPSS), Neuropathy Impairment Score (NIS) [11] and Composite Autonomic Symptom Score-31 (COMPASS-31) [12] scores for each patient. CADT is a questionnaire that evaluates the main symptoms of autonomic dysfunction observed in hATTR [9]. MRC is a numerical measurement for evaluating muscle strength [10]. NIS is specifically designed to assess polyneuropathy impairment in patients with hATTR amyloidosis and uses highly standardized, quantitative, and referenced assessments to quantify decreased muscle weakness, muscle stretch reflexes, sensory loss, and autonomic impairment [11]. COMPASS31 is a refined, internally consistent, and markedly abbreviated quantitative measure of autonomic symptoms [12].

### 2.2. Blood Tests

We performed blood tests assessing blood count, liver and renal function, thyroid function, electrolytes, proteinogram, folates, B12 vitamin, N-terminal pro-B-type natriuretic peptide (NT-proBNP), antinuclear antibody (ANA), extractable nuclear antigen (ENA) screening, creatine kinase (CK), D-dimer, C reactive protein (CRP), serum iron, serum ferritin and serum transferrin, serum glucose, kappa light chains, and kappa/lambda ratio.

### 2.3. NCS

We conducted motor NCS by assessing the median, ulnar, tibial, and peroneus nerves and sensory NCS by assessing the median, ulnar, peroneus superficialis, and sural nerves. For every nerve, we calculated sensory and motor distal latency, the amplitude of sensory action potential (SAP) or compound motor action potential (CMAP), and conduction velocity (CV).

### 2.4. Sudoscan

The electrochemical skin conductance (ESC) was evaluated in our patients with the Sudoscan device (Impeto Medical: Paris, France). All the sudomotor function tests were performed in a quiet and temperature-controlled autonomic room with an ambient temperature of 25 °C. Subjects were seated during the ESC evaluation. ESC was measured at both the hands and feet by placing the palms and soles on stainless steel electrodes for 2 min. The measurement was repeated twice, and the average ESC was calculated. Low ESC indicates a high risk of small-fiber neuropathy (SFN). An ESC of >70 µS at the feet or >60 µS at the hands indicates normal sudomotor function, while an ESC of 50–70 µS at the feet or 40–60 µS at the hands indicates moderate sudomotor dysfunction; an ESC of <50 µS at the feet or <40 µS at the hands is suggestive of severe sudomotor dysfunction [13].

### 2.5. Genetic Test

Written informed consent was collected before the genetic test for each patient. A Genilam swab test was employed for each patient. The Genilam swab test worked with a Polymerase Chain Reaction (PCR) on exons 2, 3, and 4 of the *TTR* gene (NM_000371.4) and consequent automatized sequencing (SeqStudio Flex Genetic Analyzers): this method allowed us to detect small point mutations, deletions, or insertions (<20 bp) in the above-mentioned gene. After about four weeks, every individual had received the report, which could be negative or positive. Genetic counseling was proposed for each patient.

### 2.6. Statistical Analysis

Quantitative variables are reported in raw numbers and by calculating the mean, standard deviation, and percentage.

## 3. Results

All the individuals were natives of Palazzolo Acreide, in the province of Syracuse, Sicily. We evaluated seven patients (four males, 57%). Only five patients (71%) were reported positive according to the genetic test (three females, 60%) harboring the c.242A>C (p.Glu81Ala) heterozygous variant in the *TTR* gene, generally referred to as Glu61Ala using legacy nomenclature. This variant is classified as Likely Pathogenic by ACMG/AMP 2015 guidelines, meeting criteria PS4, PM2, PM5, and PP2_supporting and reported in the ClinVar database (Variation ID:1791134) with clinical significance of probable pathogenicity.

Figure 1 describe the segregation of Glu61Ala variant in the family, as well as the distribution of phenotypes.

The mean age was 64.6 ± 10.2 years, whereas the mean age at onset was 59.4 ± 7.9 years. CADT score was 16.2 ± 5.0, NIS 9.2 ± 11.5, COMPASS31 15.6 ± 15.8, MRC 56.8 ± 4.1, and KPSS 74% ± 15.2. Four patients were classified as stage I (80%) and one as stage II (20%) in FAP. ESC in the right upper limb was 61.4 µS ± 9.7, ESC in the left upper limb 64 µS ± 7.4, ESC in the right lower limb 69.6 µS ± 10.3, and ESC in the left lower limb 72 µS ± 9.1. Two patients showed hypoevocable OTR (20%), whereas four patients showed superficial and deep sensitivity disorders (80%). All patients had a diagnosis of bilateral carpal tunnel syndrome (CTS) (100%) and polyneuropathy (100%). The sensitive nerve conduction study is reported in Table 1. Four patients showed unexplained autonomic disfunction (orthostatic hypotension, several losses of consciousness, erectile disfunction, sweating impairment, etc.) (80%) and lumbar stenosis (80%). Three patients had juvenile cardiological involvement (60%) and recent and unexplained weight loss associated with gastrointestinal signs and symptoms (diarrhea, constipation, etc.), not related to changes in dietary habits (60%). Only one patient complained about juvenile ocular problems (20%) and had a diagnosis of cardiac amyloidosis (20%). None reported bicep tendon rupture. Clinical and instrumental information for each patient is reported singularly (Table 2, Table 3 and Table 4). Currently, our patients are now on treatment with Vutrisiran, since its benefit has been demonstrated in both PN-hATTR and CM-hATTR [14,15].

### 3.1. Patient 1

Patient 1 was 72 years old, and his symptom onset occurred at 60 with two myocardial infarcts. In that circumstance, he received a diagnosis of amyloidotic cardiomyopathy through echocardiography showing a left ventricle wall thickness of 14 mm and heart scintigraphy with SPECT highlighting the presence of a grade 2 Perugini score. He suffered from dyslipidemia, lumbar stenosis with no indications of neurosurgery, bilateral CTS, and juvenile glaucoma. At the age of 70, he was diagnosed with progressive heart failure, cardiac amyloidosis, and progressive sensori-motor axonal polyneuropathy not responsive to prednisone 100 mg and azathioprine 100 mg (suspected dysimmune). Moreover, he manifested orthostatic hypotension, erectile disfunction, and unexplained loss of weight, and at the age of 71, he began to employ crutches to ambulate. During the clinical evaluation, CADT was 18, NIS 26, COMPASS31 8, KPS 60%, and MRC 50. He had a FAP stage II. ESC in the right upper limb was 50 µS, ESC in the left upper limb 52 µS, ESC in the right lower limb 58 µS, and ESC in the left lower limb 60 µS. Moreover, he showed bilateral foot drop; positive Romberg sign; decreased tactile, pain, and vibratory sensitivity in the upper and lower limbs; reduced tendon reflexes in both the upper and lower limbs; and reduced muscle strength in both the proximal and distal muscles in the lower limbs. His blood tests highlighted an increase in serum NT-proBNP (756 pg/mL, cut-off < 125 pg/mL).

### 3.2. Patient 2

Patient 2 was 74 years old, and her symptom onset occurred at 71 with a diagnosis of concentric hypertrophy of the left ventricle. She suffered from arterial hypertension, dyslipidemia, lumbar stenosis with no indications for neurosurgery, bilateral CTS, abdominal aorta stenosis, and previous total hysterectomy for gland hypertrophy. At the age of 73, she was diagnosed with progressive sensori-motor axonal polyneuropathy not responsive to prednisone 75 mg (suspected dysimmune). Moreover, she manifested orthostatic hypotension and unexplained loss of weight. During the clinical evaluation, CADT was 8, NIS 4, COMPASS-31 43, KPS 70%, and MRC 58. She had a FAP stage I. ESC in the right upper limb was 65 µS, ESC in the left upper limb 65 µS, ESC in the right lower limb 80 µS, and ESC in the left lower limb 80 µS. Moreover, she showed decreased tactile, pain, and vibratory sensitivity in right limbs, as well as reduced muscle strength in the bilateral extensor digitorum brevis muscles. Her blood tests highlighted increasing serum NT-proBNP (940 pg/mL, cut-off < 125 pg/mL), D-Dimer (1117 ng/mL, cut-off < 800 ng/mL), and total and fractionated bilirubin (2.0 mg/dL, cut-off < 1.3 mg/dL).

### 3.3. Patient 3

Patient 3 was 70 years old, and his symptom onset occurred at 62 with progressive asthenia and reduced muscle strength, as well as frequent orthostatic hypotension episodes. He suffered from mild lumbar stenosis with no indications for neurosurgery, bilateral CTS, and previously malignant lung neoplasia treated with radiotherapy. He was treated with two carotid stents in 2023 and in 2024. At the age of 66, he was diagnosed with progressive sensori-motor axonal polyneuropathy not responsive to prednisone 100 mg (suspected dysimmune). Moreover, he manifested erectile disfunction and unexplained loss of weight. During the clinical evaluation, CADT was 15, NIS 16, COMPASS31 12, KPSS 60%, and MRC 56. He had a FAP stage I. ESC in the right upper limb was 52 µS, ESC in the left upper limb 63 µS, ESC in the right lower limb 61 µS, and ESC in the left lower limb 65 µS. Moreover, he showed bilateral foot drop, reduced OTR in the upper and lower limbs, and reduced muscle strength in the bilateral tibialis anterior and extensor digitorum brevis muscles. His blood tests highlighted increases in serum NT-proBNP (681 pg/mL, cut-off < 125 pg/mL) and D-Dimer (2967 ng/mL, cut-off < 800 ng/mL), and decreases in serum iron (19 μg/dL, cut-off 33–193 μg/dL) and ferritin (15 ng/mL, cut-off 30–400 ng/mL).

### 3.4. Patient 4

Patient 4 was 53 years old, and her symptom onset occurred at 52 with progressive tingling in the feet and hands. She suffered from bilateral CTS and chronic venous insufficiency in the lower limbs. At the age of 53, she was diagnosed with progressive sensory axonal polyneuropathy. Also, she manifested with rare episodes of orthostatic hypotension. At the clinical evaluation, CADT was 20, NIS 0, COMPASS31 3, KPSS 90%, and MRC 60. She had a FAP stage I. ESC in the right upper limb was 70 µS, ESC in the left upper limb 70 µS, ESC in the right lower limb 69 µS, and ESC in the left lower limb 75 µS. Her neurological examination, echocardiogram, and blood tests were unremarkable.

### 3.5. Patient 5

Patient 5 was 54 years old, and her symptoms onset occurred at 52 with progressive tingling in the feet and hands. She suffered from bilateral CTS and cervical and lumbar stenosis. At the age of 53, she was diagnosed with progressive sensory axonal polyneuropathy. During the clinical evaluation, CADT was 20, NIS 0, COMPASS31 12, KPSS 90%, and MRC 60. She had a FAP stage I. ESC in the right upper limb was 70 µS, ESC in the left upper limb 70 µS, ESC in the right lower limb 80 µS, and ESC in the left lower limb 80 µS. Her neurological examination and echocardiogram were unremarkable. Her blood tests highlighted increasing total and fractionated bilirubin (2.3 mg/dL, cut-off < 1.3 mg/dL) and serum amylase (320 UI/L, cut-off < 225 UI/L), as well as a mild reduction in serum folates (3.1 μg/dL, cut-off 3.5–18 μg/dL).

## 4. Discussion

HATTR is a rare disorder that can affect the peripheral nervous system, the heart, or both. Several mutations in *TTR* gene are associated with CM-hATTR, while others with PN-hATTR [1,16,17]. For instance, early-onset Val30Met, Val30Ala, Gly47Arg, Gly47Glu, and Ala97Ser are usually associated with predominant neurological issues, whereas Val122Ile is almost entirely reported with predominant cardiological features [1,16,17]. Moreover, other mutations such as late-onset Val30Met, Phe64Leu, and Thr49Ala tend to cause mixed phenotypes involving both the heart and nerves [1]. One case of a rare variant (Thr60Ile) was reported to present as a mixed phenotype with CM-hATTR, polyneuropathy, and bilateral CTS, similar to the variant Glu61Ala we found [18]. However, not every *TTR* mutation is pathogenetic: in fact, notable exceptions are the super-stabilizing variant Thr119Met and the polymorphism Gly6Ser [7], which are known to be harmless. Recent THAOS data suggest that a mixed phenotype may be more common than previously thought [19]. Furthermore, over time, the same patients may experience both heart and nerve involvement, with considerable impact on the pharmacological treatment [19]. In addition, the phenotype may be influenced by ethnicity, as for the Val122Ile variant, which seems to affect the heart in African patients and the peripheral nervous system in non-Africans [20]. The pathogenetic role of Glu61Ala has never been described in the literature, but two variants affecting this codon, p. Glu61Gly and p. Glu61Lys, have been reported in individuals with transthyretin amyloidosis [21,22]. In particular, patients carrying Glu61Lys experience late onset sensory-autonomic polyneuropathy and cardiomyopathy [23,24,25,26].

Genetic penetrance in hATTR is often almost complete, although incomplete penetrance has been reported in different mutations [27]. In our description, 71% of patients reported the same mutation, suggesting a high expressivity with an elevated rate of disease development. The Glu61Ala was previously reported in only one case of CM-hATTR, with atrial fibrillation abnormal diastolic function and reduced ejection fraction, without neurological features, except for bilateral CTS [7]. Interestingly, in our study, 60% of patients showed cardiological impairment, whereas all had neurological findings, such as polyneuropathy and bilateral CTS. This suggests that the Glu61Ala variant could be connected to mixed phenotypes, differently from the previously reported case. The average age at onset in our patients was 60 years, younger compared to the data (69 years) reported in the previous case [7]. This data could suggest that earlier or later onset might affect how the disease clinically presents, as seen with the Val30Met mutation [1]; however, it is not possible to confirm this statement just considering the few elements available in the literature.

Finally, via 3D modeling (Figure 2), we found that the aminoacidic substitution of the glutamic acid with an alanine in position 61 of the mature TTR protein did not show alterations in intramolecular interactions. The substitution occurs in one of the loops connecting the β-strands of TTR protein that does not seems to be crucial for its function, but it may still affect the protein’s flexibility and interactions with amyloid fibers.

A more likely explanation is that awareness of hATTR and access to diagnostic tools are improving among general neurologists, allowing for earlier diagnosis. Moreover, the patient described by Cuddy et al. developed heart failure, like patient 1, and complained about erectile disfunction, like patients 1 and 3 [7]. The utility of Sudoscan has been demonstrated in hATTR, where its values were not only a marker of small-fiber dysfunction, but also correlated with disease severity [28,29]. Similarly, in our patients, ESC was found to be lower than the normal range in three patients (60%), indicating probable SFN. The clinical and diagnostic evaluation of the Glu61Ala mutation we have reported in this study emphasizes the importance of early screening and diagnosis, especially in the presence of red flags [8]. The limitations of our paper are the small sample size of the mutations and the lack of information on disease progression and therapeutical follow-up. More studies aiming to describe genotype–phenotype correlation for this new variant are needed to solve our doubts. Further research is needed to improve our knowledge of this new genetic variant.

## Figures and Tables

**Figure 1 jpm-15-00061-f001:**
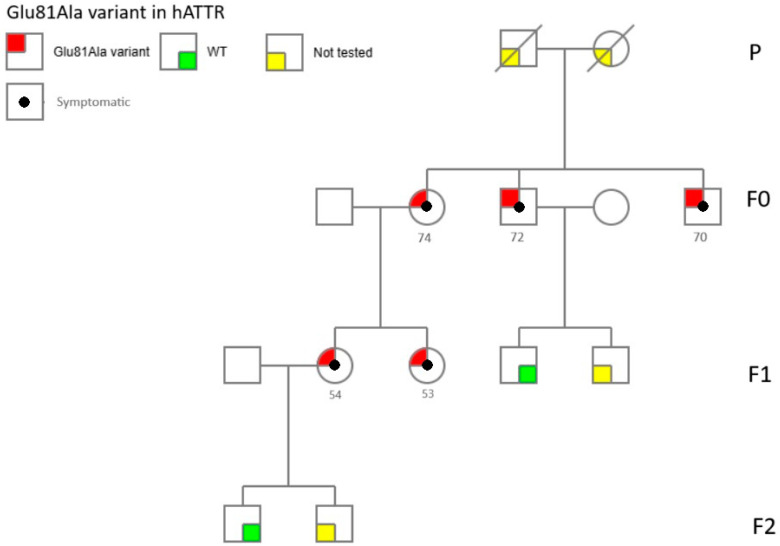
Segregation of Glu61Ala variant and clinical features in the family.

**Figure 2 jpm-15-00061-f002:**
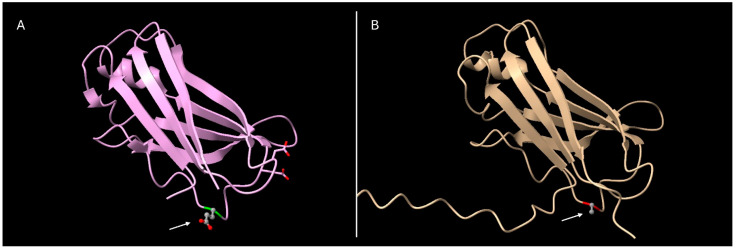
Comparison of 3D models of the wild-type (**A**) and Glu61Ala-mutated (**B**) TTR protein. Models adapted from PDB code 4TLT. White arrows indicate the amino acid position 61.

**Table 1 jpm-15-00061-t001:** Summary of sensitive nerve conduction study. SAP: sensory action potential. SCV: sensory conduction velocity.

Sensory Nerve Study	Patient 1	Patient 2	Patient 3	Patient 4	Patient 5
Ulnar nerve (right) [digit V–wrist]					
SAP (uV)	8	28	11	0	64
SCV (m/s)	49	54	50	0	56
Ulnar nerve (left) [digit V–wrist]					
SAP (uV)	8	14	32	0	59
SCV (m/s)	48	64	50	0	58
Median nerve (right) [digit III–wrist]					
SAP (uV)	0	13	0	1.6	10
SCV (m/s)	0	46	0	44	35
Median nerve (left) [digit II–wrist]					
SAP (uV)	0	12	0	0	60
SCV (m/s)	0	48	0	0	48
Sural nerve (right) [ankle–foreleg]					
SAP (uV)	0	5	0	13	8
SCV (m/s)	0	52	0	52	56
Sural nerve (left) [ankle–foreleg]					
SAP (uV)	0	5	0	22	8
SCV (m/s)	0	52	0	56	59
Superficial peroneal nerve (right) [external malleolus–dorsum of the foot]					
SAP (uV)	10	14	0	14	19
SCV (m/s)	45	60	0	58	64
Superficial peroneal nerve (left) [external malleolus–dorsum of the foot]					
SAP (uV)	9	13	24	25	23
SCV (m/s)	46	68	61	63	67

**Table 2 jpm-15-00061-t002:** Scoring features. F: female. M: male. Compound Autonomic Dysfunction Test (CADT). Medical Research Council (MRC). Familial Amyloidosis Polyneuropathy (FAP). Karnofsky Performance Status Scale (KPS). Neuropathy Impairment Score (NIS). Composite Autonomic Symptom Score-31 (COMPASS-31).

	Gender	Age	Age at Onset	CADT	NIS	COMPASS-31	MRC	KPS	FAP Stage
Patient 1	M	72	60	18	26	8	50	60	2
Patient 2	F	74	71	8	4	43	58	70	1
Patient 3	M	70	62	15	16	12	56	60	1
Patient 4	F	53	52	20	0	3	60	90	1
Patient 5	F	54	52	20	0	12	60	90	1

**Table 3 jpm-15-00061-t003:** Anamnestic and clinical features. F: female. M: male. CTS: Carpal Tunnel Syndrome.

	Gender	Age	Age at Onset	Bilateral CTS	Polyneuropathy	Cardiological Involvement	Unexplained Loss of Weight	Autonomic Dysfunction	Lumbar Stenosis	Juvenile Ocular Disorders
Patient 1	M	72	60	Y	Y	Y	Y	Y	Y	Y
Patient 2	F	74	71	Y	Y	Y	Y	Y	Y	N
Patient 3	M	70	62	Y	Y	Y	Y	Y	Y	N
Patient 4	F	53	52	Y	Y	N	N	Y	N	N
Patient 5	F	54	52	Y	Y	N	N	N	Y	N

**Table 4 jpm-15-00061-t004:** Sudoscan features. F: female. M: male. Electrochemical skin conductance (ESC). An ESC of >70 µS at feet or >60 µS at the hands indicates normal sudomotor function, while an ESC of 50–70 µS at the feet or 40–60 µS at the hands indicates moderate sudomotor dysfunction; ESC of <50 µS at the feet or <40 µS at the hands is suggestive of severe sudomotor dysfunction. Values marked with “*” indicate impaired sudomotor function upon Sudoscan.

	Gender	Age	Age at Onset	Right Upper Limb ESC	Left Upper Limb ESC	Right Lower Limb ESC	Left Lower Limb ESC
Patient 1	M	72	60	50 *	52 *	58 *	60 *
Patient 2	F	74	71	65	65	80	80
Patient 3	M	70	62	52 *	63	61 *	65 *
Patient 4	F	53	52	70	70	69 *	75
Patient 5	F	54	52	70	70	80	80

## Data Availability

Data are available from the corresponding author upon reasonable request.

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
