# Peer review of "Expanding the Genetic and Clinical Spectrum of Hereditary Transthyretin Amyloidosis: The Glu61Ala Variant"

_jpm, 2025, doi:10.3390/jpm15020061_

Round 1
Reviewer 1 Report
Comments and Suggestions for Authors
This is very important work and the authors have performed a thorough evaluation of the patients. I would recommend mostly improving the quality of the English language as it decreases the overall value of the manuscript.
Few minor comments
Introduction:
- Avoid the word death better to use the word mortality
- Reporting prevalence page 2 first line:
Please report either as a percentage or or as the number of cases per 10,000 or 100,000 people.
“10186 people, with a range between 5000 and 38000(2).”
- Change the word “peculiar” in “…and every mutation shows a peculiar phenotype…” I don’t think this is the word you are looking for here, perhaps distinct or unique?
- Please revise the English language, particularly in the discussion
Comments on the Quality of English LanguageThis is very important work and the authors have performed a thorough evaluation of the patients. I would recommend mostly improving the quality of the English language as it decreases the overall value of the manuscript.
Author Response
Dear Editors and Reviewers
Thanks for your comments. We would like to submit our revised version of the manuscript for possible publication in your journal.
Reviewer 1
A: This is very important work and the authors have performed a thorough evaluation of the patients. I would recommend mostly improving the quality of the English language as it decreases the overall value of the manuscript.
Few minor comments
Introduction:
- Avoid the word death better to use the word mortality
- Reporting prevalence page 2 first line:
Please report either as a percentage or or as the number of cases per 10,000 or 100,000 people.
“10186 people, with a range between 5000 and 38000(2).”
- Change the word “peculiar” in “…and every mutation shows a peculiar phenotype…” I don’t think this is the word you are looking for here, perhaps distinct or unique?
- Please revise the English language, particularly in the discussion
R: We thank the reviewer for her/his precious suggestions. We modified the manuscript according to the comments. A careful revision from a native-speaking English was provided for the whole manuscript.
Hoping in positive feedback we look forward to hearing from you soon.
Kind regards,
Vincenzo Di Stefano
Reviewer 2 Report
Comments and Suggestions for Authors
This paper described a novel mutation in TTR gene, the Glu61Ala variant, which had been previously reported only in one case of cardiac phenotype, and its clinical findings. Materials and Methods. One individual affected by chronic idiopathic polyneuropathy and a major red flag for hATTR underwent genetic saliva testing to look for mutations in the TTR gene. Then, his relatives were subjected to the same test. We assessed the anamnestic profile, general and neurological examination, blood tests, nerve conduction studies (NCS), electrocardiogram and Sudoscan for each patient. Written informed consent has been acquired for every patient. Results. Among 7 patients screened, only 5 patients carried the Glu61Ala variant (71%). The mean age was 64.6 ± 10.2 years, whereas the mean age at onset was 59.4 ± 7.9 years. In our study, three patients (60%) showed a mixed phenotype, whereas two of them (40%) showed a neurological phenotype. Discussion. The Glu61Ala variant was reported only in one case with a cardiological phenotype, but our patients showed both neurological and cardiological involvement. Further studies are needed to improve knowledge of this genetic variant.
Comments on the manuscript "Expanding the genetic and clinical spectrum of Hereditary Transthyretin Amyloidosis: the Glu61Ala variant" by Christian Messina and colleagues.
Interesting finding.
My minor comments:
Did patients 2-5 also underwent bone scintigraphy, as echo can be (false) negative?
Author Response
Dear Editors and Reviewers
Thanks for your comments. We would like to submit our revised version of the manuscript for possible publication in your journal.
Reviewer 2
This paper described a novel mutation in TTR gene, the Glu61Ala variant, which had been previously reported only in one case of cardiac phenotype, and its clinical findings. Materials and Methods. One individual affected by chronic idiopathic polyneuropathy and a major red flag for hATTR underwent genetic saliva testing to look for mutations in the TTR gene. Then, his relatives were subjected to the same test. We assessed the anamnestic profile, general and neurological examination, blood tests, nerve conduction studies (NCS), electrocardiogram and Sudoscan for each patient. Written informed consent has been acquired for every patient. Results. Among 7 patients screened, only 5 patients carried the Glu61Ala variant (71%). The mean age was 64.6 ± 10.2 years, whereas the mean age at onset was 59.4 ± 7.9 years. In our study, three patients (60%) showed a mixed phenotype, whereas two of them (40%) showed a neurological phenotype. Discussion. The Glu61Ala variant was reported only in one case with a cardiological phenotype, but our patients showed both neurological and cardiological involvement. Further studies are needed to improve knowledge of this genetic variant.
Comments on the manuscript "Expanding the genetic and clinical spectrum of Hereditary Transthyretin Amyloidosis: the Glu61Ala variant" by Christian Messina and colleagues.
Interesting finding.
My minor comments:
Did patients 2-5 also underwent bone scintigraphy, as echo can be (false) negative?"
R: We thank the reviewer for her/his precious suggestions. Scintigraphy was performed only for patient 1. We agree with the reviewer on the low predictive value of echo, but it was not performed in all cases as the patients refused. However, the strong positivity of scintigraphy in patient 1 is enough to confirm that this mutation is not silent on bone tracers for scintigraphy in Glu61Ala variant as happens for Phe84Leu patients.
Hoping in positive feedback we look forward to hearing from you soon.
Kind regards,
Vincenzo Di Stefano